# High Prevalence of Novel Sequence Types in *Streptococcus pneumoniae* That Caused Invasive Diseases in Kuwait in 2018

**DOI:** 10.3390/microorganisms12010225

**Published:** 2024-01-22

**Authors:** Eiman Mokaddas, Mohammad Asadzadeh, Shabeera Syed, M. John Albert

**Affiliations:** Department of Microbiology, College of Medicine, Kuwait University, P.O. Box 24923, Safat 13110, Kuwait; eiman.mokaddas@ku.edu.kw (E.M.); mohammad.assadzadeh@ku.edu.kw (M.A.); shabeera.rahim@ku.edu.kw (S.S.)

**Keywords:** *S. pneumoniae*, MLST, sequence type, serotype, phylogenetic analysis, Kuwait

## Abstract

Background: Multilocus sequence typing (MLST) is used to gain insight into the population genetics of bacteria in the form of sequence type (ST). MLST has been used to study the evolution and spread of virulent clones of *Streptococcus pneumoniae* in many parts of the world. Such data for *S. pneumoniae* are lacking for the countries of the Arabian Peninsula, including Kuwait. Methods: We determined the STs of all 31 strains of *S. pneumoniae* from invasive diseases received at a reference laboratory from various health centers in Kuwait during 2018 by MLST. The relationship among the isolates was determined by phylogenetic analysis. We also determined the serotypes by Quellung reaction, and antimicrobial susceptibility by Etest, against 15 antibiotics belonging to 10 classes. Results: There were 28 STs among the 31 isolates, of which 14 were new STs (45.2%) and 5 were rare STs (16.1%). Phylogenetic analysis revealed that 26 isolates (83.9%) were unrelated singletons, and the Kuwaiti isolates were related to those from neighboring countries whose information was gleaned from unpublished data available at the PubMLST website. Many of our isolates were resistant to penicillin, erythromycin, and azithromycin, and some were multidrug-resistant. Virulent serotype 8-ST53, and serotype 19A with new STs, were detected. Conclusions: Our study detected an unusually large number of novel STs, which may indicate that Kuwait provides a milieu for the evolution of novel STs. Novel STs may arise due to recombination and can result in capsular switching. This can impact the effect of vaccination programs on the burden of invasive pneumococcal disease. This first report from the Arabian Peninsula justifies the continuous monitoring of *S. pneumoniae* STs for the possible evolution of new virulent clones and capsular switching.

## 1. Introduction

*Streptococcus pneumoniae* is the causative agent of invasive diseases such as sepsis, pneumonia, and meningitis worldwide and contributes to a significant morbidity and mortality especially among young children and the elderly [1,2]. Invasive *S. pneumoniae* infections are common in Kuwait [3,4]. Currently, there are about 100 serotypes of *S. pneumoniae* [5]. The changing prevalence of serotypes following vaccination [6], and the emergence of antibiotic resistance [7], necessitate the need for molecular epidemiology and the surveillance of *S. pneumoniae.* Several molecular typing methods have been developed to type *S. pneumoniae* [8]. Among these typing methods, multilocus sequence typing (MLST) can provide an adequate discrimination of strains for local and global epidemiology [8]. MLST can provide insights into the evolutionary changes in circulating clones and the emergence of new clones [9] and serotypes [10]. It is also useful to ascertain the spread of virulent international clones [11], including multidrug-resistant isolates [12]. There are no data on the sequence types (STs, determined by MLST) of *S. pneumoniae* circulating in Kuwait, a member of the Gulf Cooperation Council (GCC) countries of the Middle East. There are also no published data in journals on the prevalence of STs of *S. pneumoniae* among the other GCC countries (Saudi Arabia, Qatar, Oman, and the United Arab Emirates), even though some isolates have been submitted to the PUBMLST database (https://pubmlst.org/organisms/streptococcus-pneumoniae) (accessed on 3 October 2023) from Saudi Arabia, Qatar, and Oman. Therefore, we determined the STs of *S. pneumoniae* causing invasive infections during 2018 in Kuwait and their relationship to those of *S. pneumoniae* from the above three GCC countries, and the neighboring countries, Iraq and Iran. This report is the first one from the region covering the GCC countries.

## 2. Materials and Methods

### 2.1. S. pneumoniae Isolates

Pneumococcal isolates originated from sterile site specimens—blood and cerebrospinal fluid (CSF)—of patients seeking care at the general hospitals, tertiary-care hospitals, and polyclinics in Kuwait during 2018. A single isolate per patient was studied. If a patient had isolates from both blood and CSF at the same time, the CSF isolate was chosen. The isolates were sent on blood agar plates to the pneumococcal reference laboratory at the College of Medicine, Kuwait University, Jabriya, Kuwait. The isolates were reconfirmed as *Streptococcus pneumoniae* by positive tests for α-hemolysis on blood agar (Oxoid, Basingstoke, UK), optochin susceptibility (Oxoid), and bile solubility (Sigma, St. Louis, MO, USA). The isolates were stored in blood glycerol broth (2.5 g nutrient broth and 16.8 mL glycerol, made up to 95 mL with distilled water, and sterilized and supplemented with 5 mL sterile sheep blood) [13] at −70 °C for further study. Limited demographic data of patients, including age and gender, were recorded.

Serotyping of the isolates was performed by Quellung reaction with rabbit polyclonal antisera (Statens Serum Institute, Copenhagen, Denmark) in the Danish chessboard typing system [14]. The Quellung reaction was performed by mixing a 15 µL suspension of the bacteria in saline with an equal volume of the undiluted *S. pneumoniae* antiserum on a microscopic slide and observing the capsular swelling under a phase-contrast microscope. Full serotyping of all isolates was carried out according to the instructions of the supplier of the antisera. For a full serotyping, the following antisera (Statens Serum Institute) were used: omni serum (reacting with all known serotypes), pool antisera (A to I and P to T, for typing or grouping most pneumococci), typing sera (reacting with a single serotype), group sera (reacting with all the types within one group), and factor sera (for differential typing within a group).

### 2.2. DNA Preparation

*S. pneumoniae* was grown on tryptone soya agar (Oxoid) or brain heart infusion agar (Oxoid) supplemented with 5% sheep blood in a 5% CO_2_ atmosphere at 37 °C for 20 h. The genomic DNA was extracted using the Chelex-100 fast procedure. A loop of bacterial colonies from an agar plate was suspended in 1.0 mL of sterile, deionized water in a microcentrifuge tube containing 50 mg of Chelex-100 (Sigma), heated at 95 °C for 10 min, and then centrifuged in a Spectrafuge 24D (Labnet, Edison, NJ, USA) at 15.6× *g* for 5 min. The supernatant was used as the source of DNA [15].

### 2.3. PCR Assays

The internal fragments of seven housekeeping genes (*aroE*, *gdh*, *gki, recP*, *spi*, *xpt*, and *ddl*) [16] were amplified by PCR. The primers used were those described by Enright and Spratt [16] and were purchased from Eurogentec (Seraing, Belgium). The primers used for PCR amplification of genes were for *aroE* (aroeF 5′-GCCTTTGAGGCGACAGC-3′ and aroeER 5′-TGCAGTTCA(G/A)AAACAT(A/T)TTCTAA-3′), *gdh* (gdhF 5′-ATGGACAAACCAGC(G/A/T/C) AG(C/T)TT-3′ and gdhR 5′-GCTTGAGGTCCCAT(G/A)CT(G/A/T/C)CC-3′), *gki* (gkiF 5′-GGCATTGGAATGGGATCACC-3′ and gkiR 5′-TCTCCCGCAGCTGACAC-3′), *recP* (recPF 5′-GCCAACTCAGGTCATCCAGG-3′ and recPR 5′-TGCAACCGTAGCATTGTAAC-3′), *spi* (spiF 5′-TTATTCCTCCTGATTCTGTC-3′ and spiR 5′-GTGATTGGCCAGAAGCGGAA-3′), *xpt* (xptF 5′-TTATTAGAAGAGCGCATCCT-3′ and xptR 5′-AGATCTGCCTCCTTAAATAC-3′) and *ddl* (ddlF 5′-TGC(C/T)CAAGTTCCTTATGTGG-3′ and ddlR 5′-CACTGGGT(G/A)AAACC(A/T)GGCAT-3′). As the sources of PCR reagents and cycling conditions were different from those used by Enright and Spratt [10], the assay is described below. The total reaction volume was 25 µL, which consisted of 12.5 µL ReadyMix Taq PCR Reaction Mix with MgCl2 (Sigma), 8.5 µL water, 1 µL (5 pmol) each of forward and reverse primers, and 2 µL (50 ng) DNA template. The reaction consisted of initial denaturation at 94 °C for 4 min, followed by 40 cycles of 94 °C/1 min (denaturation); 50 °C (for *aroE*), 56 °C (for *spi* and *xpt*), and 60 °C (for *gdh*, *gki*, *recP* and *ddl*)/2 min (annealing); 74 °C/2 min (extension); and a final extension at 74 °C/5 min. The PCR product was run on a 1.5% agarose gel containing ethidium bromide (Promega, Madison, WI, USA) in Tris-Borate-EDTA (TBE) buffer (Promega) at 120 volts for 30 min.

### 2.4. MLST Determination

The PCR product was purified with a QIAquick PCR Purification Kit (Qiagen, Germantown, MD, USA) according to the supplied instructions. Both strands were sequenced using the BigDye terminator DNA sequencing kit (ABI Prism BigDye terminator v3.1) (Applied Biosystems, Norwalk, CT, USA) using the same primers that were used for amplification. The sequencing products were processed and analyzed using the DNA sequencer (ABI Prism 3130xl Genetic Analyzer) (Applied Biosystems) according to the manufacturer’s instructions. The allele profile of each sample was investigated with the forward and reverse sequences using Clustal omega (https://www.ebi.ac.uk/Tools/msa/clustalo/). Sequence types (STs) were determined by submitting the allele sequences of the seven genes to the PubMLST database (https://pubmlst.org/organisms/streptococcus-pneumoniae), accessed on 15 May 2023. For indeterminate STs, novel STs were assigned by the curators of the database.

Based on the allele number for the seven gene fragments for each isolate, a dendrogram was constructed using BioNumerics v7.5 software (Applied Maths, Sint-Martens-Latem, Belgium) and standard unweighted pair group method with arithmetic mean (UPGMA) settings. The genetic relationship among the genotypes was studied by constructing a minimum spanning tree using the Bionumerics software, which predicts the relationships among the isolates and records two isolates as more closely related when six of the seven loci are identical [17]. The genetic relationship between the Kuwaiti isolates and the isolates from neighboring countries (Saudi Arabia, Qatar, Oman, Iraq, and Iran) was also investigated.

### 2.5. Antibiotic Susceptibility Testing

Antibiotic susceptibility testing was carried out using Etest strips (bioMerieux, Marcy-l’Etoile, France) on Mueller–Hinton sheep blood agar (Oxoid) and culture incubation was performed at 37 °C in a 5% CO_2_ atmosphere for 20 h. Susceptibility testing was carried out against penicillins (penicillin, amoxicillin, amoxicillin–clavulanic acid), macrolides (erythromycin, azithromycin), cephalosporins (cefuroxime, cefotaxime, ceftriaxone), tetracycline, sulfonamide (trimethoprim–sulfamethoxazole), phenicol (chloramphenicol), glycopeptide (vancomycin), fluoroquinolones (levofloxacin, moxifloxacin), carbapenem (meropenem), and lincosamide (clindamycin). The interpretative criteria were according to the CLSI guidelines [18]. Penicillin MIC susceptibility breakpoints for meningeal isolates are ≤0.06 mg/L (susceptible), and ≥0.12 mg/L (resistant). The breakpoints for non-meningeal isolates are ≤2 mg/L (susceptible), 4 mg/L (intermediate susceptible) and ≥2 mg/L (resistant). The breakpoints for two cephalosporins (cefotaxime and ceftriaxone) for meningeal versus non-meningeal isolates are ≤0.5 mg/L (susceptible), 1 mg/L (intermediate susceptible), and ≥2 mg/L (resistant) versus ≤1 mg/L (susceptible), 2 mg/L (intermediate susceptible), and ≥4 mg/L (resistant). With each round of testing, the quality control strain of *S. pneumoniae* ATCC strain 49619 was also tested. The results of susceptibility testing were accepted if the results of the control strain were within the published limits.

## 3. Results

The demographics of the 31 patients, the nature of specimens, the hospitals where treatment was provided, and the serotypes and STs of *S. pneumoniae* are shown in Table 1. All but four isolates were cultured from blood and the four isolates were cultured from CSF. The patients were spread over various ages of either gender. The patients were treated at five hospitals. The information regarding patient identity, clinical data, admission ward, and nationality was not available to us. There were 22 serotypes among the 31 isolates. Serotypes, 9N, 15A, 19A, 8, 22A, 23A, 29, 6C/6D, and 23B were represented by two isolates each. The other 13 serotypes were represented by single isolates.

The 31 isolates were represented by 28 STs. There were 14 new STs (17466, 17468, 17470, 17471, 17474, 17476, 17479, 17481, 17483, 17484, 17485, 17488, 17771, 17776) among the 31 isolates (45.2%). Two isolates had the same new ST, 17771. Thus, new STs accounted for 15/31 (48.4%) of the isolates (Table 1). Three isolates had novel alleles for *recP* gene. Other new STs had new combinations of alleles (Table 2).

Duplicate isolates of serotypes 9N, 15A, 23B, 8, 22A, and 29, all originating at different hospitals, belonged to two different STs, respectively. Duplicate isolates of serotypes 23A and 6C/6D, with duplicate isolates originating at different hospitals, belonged to the same STs, respectively (Table 1).

The antibiotic susceptibility profiles of the isolates are shown in Appendix A. Five isolates were susceptible to all antibiotics. Two isolates were resistant to penicillin. Many isolates were resistant to erythromycin and azithromycin. All four isolates cultured from CSF were multidrug-resistant, being resistant to three or more classes of antibiotics. Three blood isolates (1949, 2018, 2028) were multidrug-resistant. There was no distinct pattern of resistance among the isolates with novel STs. 

The genetic relationships among the *S. pneumoniae* isolates from Kuwait were assessed by constructing an unrooted phylogenetic tree using the MLST data. The dendrogram (Figure 1) showed that 26 of the 31 isolates (83.9%) were unrelated singletons, each belonging to a single ST. The remaining five isolates were divided into two clusters, ST1390 and ST17771, found in two hospitals. ST1390 was found in *S. pneumoniae* serotypes 6C and 6C/6D, while ST17771 was found in serotype 23A (Table 1). MLST analysis revealed that patients in nearly all the clusters were distributed across various hospitals in Kuwait, and no clear association was found between specific STs and hospitals (Figure 2).

The genetic relationship of the population structure of *S. pneumoniae* isolates from Kuwait with those from five neighboring countries (Saudi Arabia, Iraq, Iran, Qatar, and Oman) was explored using the data from the *S. pneumoniae* MLST database (Figure 3). The Kuwaiti isolates clustered alongside those from these neighboring countries, with no distinct clustering with any country (Figure 3).

## 4. Discussion

A high percentage of *S. pneumoniae* isolates belonged to novel STs in our study. This prevalence is unusually high. Studies in other geographical locations have encountered novel STs. However, the prevalence of novel STs in these studies was low. For example, it was 9.3–11% in China [19,20], 3.3% in Turkey [21], 6.5% in Portugal [22], and 8.7% in Argentina [23]. This shows an unusually high evolution of new STs in Kuwait. Genomic variation in *S. pneumoniae* occurs through two mechanisms: spontaneous mutation (two–four novel mutations per genome per year [24]) and recombination. *S. pneumoniae* is a naturally competent organism and easily picks up DNA from the surroundings by transformation [25]. Nearly 90% of all polymorphisms in the pneumococcus have been introduced through recombinational exchanges [26]. Every recombination event gives rise to an average of 72 mutations per isolate [26,27]. Kuwait has a population of about 4.3 million people, and 70% is an expatriate population from many countries of Asia, Africa, Europe, and North America (www.worldpopulationreview.com/countries/kuwait-population, accessed on 29 October 2023). Therefore, Kuwait provides a favorable milieu for the mixing of strains and evolution of new strains. Other factors driving mutations are host immunity through vaccination [4] and antibiotic pressure [28]. Recombination taking place in a structural protein, or a vaccine candidate structure, can alter the virulence of the strain. This can make the strain resistant to the host immune response and increase antibiotic resistance [29]. Pneumococcal isolates can undergo capsular switching whereby the serotype of a strain changes due to an alteration in the capsular biosynthesis locus or by a genetic recombination. An increase in the prevalence of capsule-switched variants can result in serotype replacement. Such a replacement of vaccine serotypes by non-vaccine serotypes has the potential to reduce the impact of vaccination on the burden of invasive pneumococcal diseases [29]. It has been suggested that disease potential is associated not only with serotype but also with clonal type determined by MLST [30]. About 30% of the pneumococcal gene content is made up of accessory genes which encode virulence genes, antimicrobial resistance, etc. A correlation has been shown between MLST and the accessory gene content [31]. It may become possible in future to predict the virulence of a strain based on its MLST profile.

Among the previously reported 14 STs (Table 1), 5 STs (ST53, ST989, ST230, ST473, ST618) have a worldwide prevalence. Four of these have been previously found in other GCC countries, except ST473. Since ST473 is found in our study, ours is the first study to report this ST from a GCC country. Four additional STs (ST1390, ST1373, ST1233, ST517) have a moderate prevalence. All except ST1233 are reported for the first time in our region. The remaining five STs are less prevalent in the world, with ST2685 and ST8959 having been found previously in GCC countries: ST7340 being found in Spain; ST1982 in the United States of America; and ST17348 in the United Kingdom, as well as in our study ((https://pubmlst.org/organisms/streptococcus-pneumoniae) (accessed on 3 October 2023)).

The genetic relationships of our isolates were explored by constructing a dendrogram. This suggested that most of our isolates were singletons and therefore unrelated. We also could not find any association between specific STs and hospitals. This could be due to the low number of isolates that we studied. A comparison with regional isolates showed Kuwaiti isolates clustering with the isolates from these countries.

In our study, there were no dominant STs found, probably due to the low number of isolates studied. In some studies, some STs were found to be dominant. For example, the dominant STs were ST320, ST271, ST81, ST876, and ST3173 in China [19]; ST179, ST2918, ST386, and ST3772 in Tunisia [32]; ST320 and ST13223 in Vietnam [33]; ST618 and ST3081 in the Gambia [34]; and ST306, ST191, ST989, and ST180 in Spain [35].

The same serotypes of *S. pneumoniae* exhibited the same ST or different STs in our study. This has been observed in other studies as well [19,36,37].

Some of our isolates were resistant to penicillin, macrolides or were multidrug-resistant. Resistance to many antibiotics in *S. pneumoniae* has been reported in many parts of the world (reviewed in reference [38]), including from Kuwait [28].

Our data do not fulfill the criteria defined by The Pneumococcal Molecular Epidemiology Network (PMEN) for the inclusion of our isolates in international clones [39]. Among the many criteria, we have not established the presence of a clone in Kuwait for many years. Even so, some serotypes isolated in our study are of interest. We had two isolates of serotype 8. One isolate belonged to ST53 and was susceptible to all the antibiotics tested. The incidence of invasive disease, due to this serotype 8-ST53 susceptible clone, has increased after the introduction of the PV13 vaccine in Denmark [40]. We had two isolates of serotype 19A belonging to two novel STs. One isolate was multidrug-resistant and the other one was intermediate-resistant to trimethoprim–sulfamethoxazole. The incidence of invasive disease with the serotype 19A belonging to multiple STs and variable resistance has increased in many countries after the introduction of PCV7 and PCV10 conjugate vaccines [41,42]. There could be other serotypes that have epidemic potential [43]. It would be interesting to watch how the clones related to the relevant serotypes evolve in Kuwait.

The main limitation of our study is that we analyzed a relatively small number of isolates. However, we included all the isolates collected during 2018. The polysaccharide pneumococcal vaccine, PCV23, and the conjugate pneumococcal vaccine, PCV13, have been in use in Kuwait for many years now [4]. The effect of vaccination on a small population has contributed to the relatively small number of invasive pneumococcal infections and isolates. Another limitation is that we did not have the detailed demographic and clinical information on the patients for an in-depth analysis of the data.

## 5. Conclusions

The current study, even though conducted using a small number of isolates, has yielded valuable information on circulating STs of *S. pneumoniae* in Kuwait. The study has found numerous novel STs as well as some rare STs. This study is the first one from a region representing the GCC countries and contributes to the world literature on the population genetics of *S. pneumoniae.* This study justifies the continuous monitoring of the population genetics of *S. pneumoniae* for the possible evolution of virulent international clones and capsular switching.

## Figures and Tables

**Figure 1 microorganisms-12-00225-f001:**
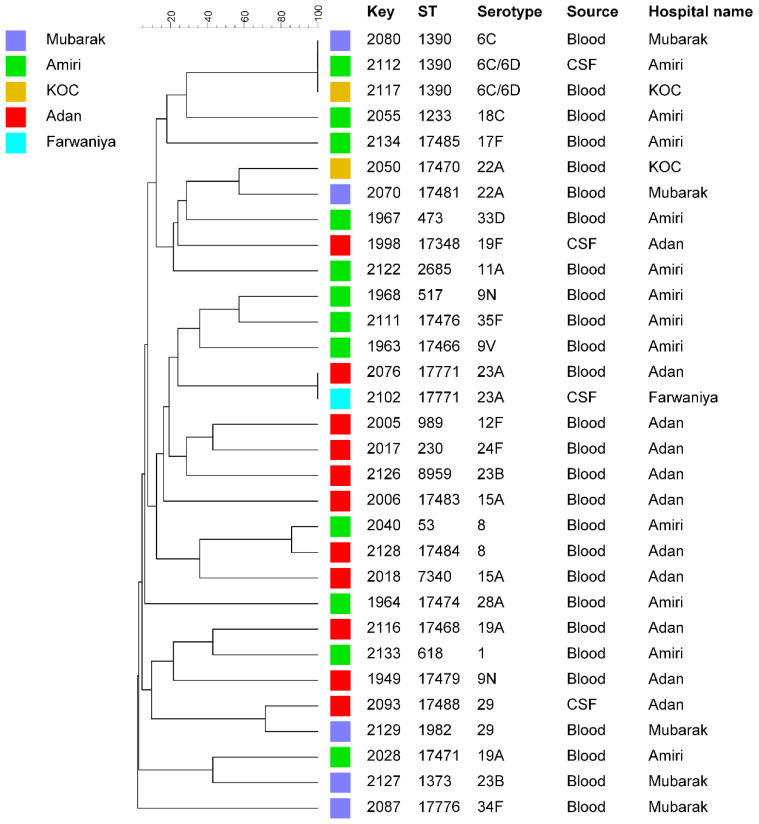
UPGMA tree based on MLST sequence data from 31 *S. pneumoniae* isolate. Similarity is presented in percentage using the scale bar in the upper left corner. The columns from left to right include, isolate number (key), MLST-based sequence type (ST), serotype, clinical source, and the hospital name where the isolate originated.

**Figure 2 microorganisms-12-00225-f002:**
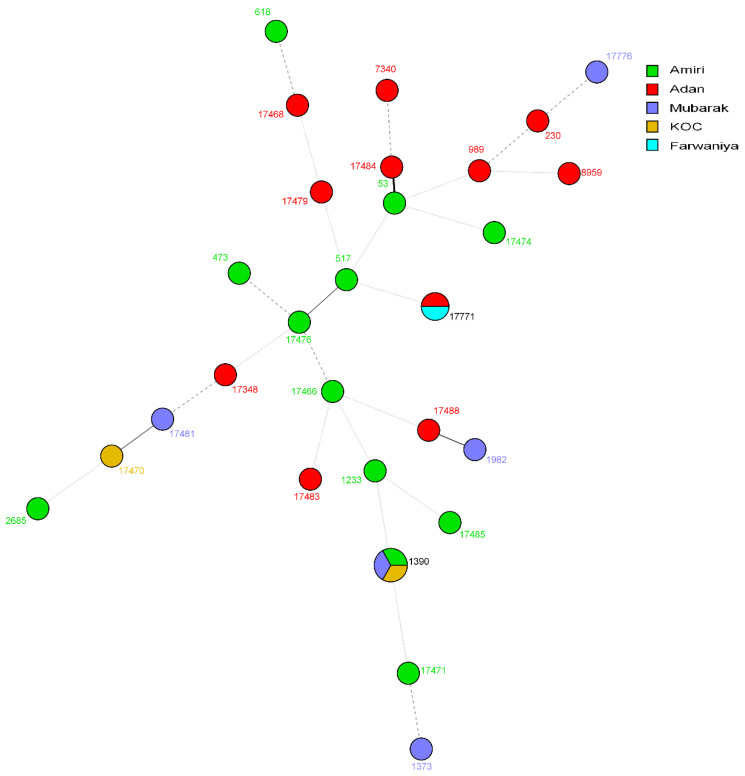
Minimum spanning tree showing the genotypic relationship among *S. pneumoniae* isolated in different hospitals in Kuwait. Each circle corresponds to a unique genotype, and lines between the circles represent relative distance between isolates. The sizes of the circles correspond to the number of isolates of the same genotype (ST). Connecting lines correspond to the number of allelic differences between genotypes, with a solid thick line connecting genotypes that differ in one locus, a solid thin line connecting genotypes that differ in two-three loci, a dashed line connecting genotypes that differ in four loci, and a dotted line connecting genotypes that differ in more than four loci.

**Figure 3 microorganisms-12-00225-f003:**
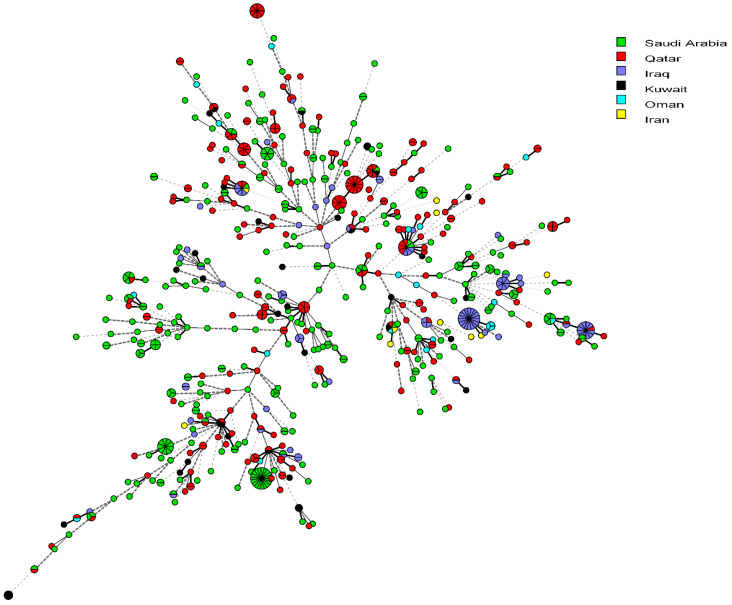
Minimum spanning tree showing the relationship of *S. pneumoniae* from Kuwait with 684 isolates from neighboring countries (Saudi Arabia, Qatar, Oman, Iraq, and Iran) available from the MLST website as of 3 October 2023. Each circle corresponds to a unique genotype, and lines between the circles represent relative distance between isolates. The sizes of the circles correspond to the number of isolates of the same genotype (ST). Connecting lines correspond to the number of allelic differences between genotypes, with a solid thick line connecting genotypes that differ in one locus, a solid thin line connecting genotypes that differ in two-three loci, a dashed line connecting genotypes that differ in four loci, and a dotted line connecting genotypes that differ in more than four loci.

**Table 1 microorganisms-12-00225-t001:** Patient and the isolate information for *S. pneumoniae.*

Isolate No.	Specimen	Patient		Hospital	Serotype	Sequence Type
		Age (y/m/d)	Gender			(ST)
1949	Blood	11 m	M	Adan	9N	17479 *
1963	Blood	53 y	M	Amiri	9V	17466 *
1964	Blood	2 y	M	Amiri	28A	17474 *
1967	Blood	78 y	F	Amiri	33D	473
1968	Blood	38 y	F	Amiri	9N	517
1998	CSF	38 y	M	Adan	19F	17348
2005	Blood	10 d	F	Adan	12F	989
2006	Blood	21 y	F	Adan	15A	17483 *
2017	Blood	2 y	F	Adan	24F	230
2018	Blood	11 y	M	Adan	15A	7340
2028	Blood	25 y	F	Amiri	19A	17471 *
2040	Blood	32 y	F	Amiri	8	53
2050	Blood	71 y	F	KOC	22A	17470 *
2055	Blood	60 y	M	Amiri	18C	1233
2070	Blood	21 y	M	Mubarak	22A	17481 *
2076	Blood	29 y	F	Adan	23A	17771 *
2080	Blood	59 y	M	Mubarak	6C	1390
2087	Blood	4 y	M	Mubarak	34F	17776 *
2093	CSF	64 y	F	Adan	29	17488 *
2102	CSF	2 y	M	Farwaniya	23A	17771 *
2111	Blood	85 y	F	Amiri	35F	17476 *
2112	CSF	50 y	M	Amiri	6C/6D	1390
2116	Blood	32 y	M	Adan	19A	17468 *
2117	Blood	75 y	M	KOC	6C/6D	1390
2122	Blood	69 y	M	Amiri	11A	2685
2126	Blood	3 m	M	Adan	23B	8959
2127	Blood	4 y	F	Mubarak	23B	1373
2128	Blood	6 d	F	Adan	8	17484 *
2129	Blood	9 m	M	Mubarak	29	1982
2133	Blood	49 y	M	Amiri	1	618
2134	Blood	72 y	M	Amiri	17F	17485 *

* New ST.

**Table 2 microorganisms-12-00225-t002:** Gene allele information found for new STs of *S. pneumoniae* by multilocus sequence typing.

Specimen No.	Allele for Gene							New ST
	*aroE*	*gdh*	*gki*	*recP*	*spi*	*xpt*	*ddl*	
1949	2	720	174	4	7	435	5	17479
1963	15	17	4	16	6	1	271	17466
1964	16	44	1	4	9	70	17	17474
2006	4	41	47	16	6	14	2	17483
2028	8	13	8	6	461	6	18	17471
2050	118	16	751	4	15	1	9	17470
2070	7	16	751	1	15	1	14	17481
2076	54	5	9	546 *	6	273	8	17771
2087	5	19	2	547 *	121	22	265	17776
2093	296	12	4	44	14	77	97	17488
2102	54	5	9	546 *	6	273	8	17771
2111	7	5	4	4	6	1	79	17476
2116	2	18	4	1	7	232	9	17468
2128	2	5	1	11	16	3	271	17484
2134	10	11	426	5	6	88	9	17485

* New allele.

## Data Availability

The data presented in this study are available on request from the corresponding author.

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
