# Peer review of "High Prevalence of Novel Sequence Types in Streptococcus pneumoniae That Caused Invasive Diseases in Kuwait in 2018"

_microorganisms, 2024, doi:10.3390/microorganisms12010225_

Round 1

Reviewer 1 Report

Comments and Suggestions for Authors

In the manuscript “High Prevalence of Novel Sequence Types of Streptococcus pneumoniae Causing Invasive Diseases in Kuwait in 2018”, Mokaddas investigated the sequence types (STs) of S. pneumoniae strains causing invasive diseases in Kuwait during 2018. The study employs Multilocus Sequence Typing (MLST) and analyzes 31 strains, showing 28 different STs, with a significant proportion being novel. The antibiotic resistance patterns of these strains were also analyzed. The manuscript does indeed have an issue with the arrangement and formatting, as observed in the section discussing Figure 3. The heading "4. Discussion" appears inappropriately within the description of Figure 3, indicating a formatting error. This placement disrupts the flow and organization of the paper, making it difficult for readers to follow the logical progression of sections. To correct this, the "Discussion" section should be clearly separated and positioned after the description of Figure 3, ensuring it stands as an independent section. This will improve the readability and structural coherence of the manuscript. Additionally, a thorough review of the entire manuscript for similar formatting inconsistencies, especially regarding headings and font sizes, is recommended to ensure uniformity and clarity throughout the document. In addition, the authors should analyze the data, focus on the high prevalence of novel STs, and explore the implications of this finding in the context of regional public health and S. pneumoniae evolution. Overall, the manuscript in current state did not be prepared to be an acceptable publication.

Author Response

Reply in the enclosed file

Reviewer 2 Report

Comments and Suggestions for Authors

The authors presented the results of a study on the diversity of sequence types of Streptococcus pneumoniae in Kuwait. The obtained data will be interesting to both epidemiologists and microbiologists. However, As mentioned by the authors, a very small number of bacterial isolates were studied. For this reason, I would recommend adjusting the title of the article. For example, "Multilocus sequence typing analysis of Streptococcus pneumoniae in Kuwait (preliminary results)".

It seems to me that the caption of figure 2 contain sentences of the main text. Please check it.

The head of section Discussion is included in the caption of figure 3.

Author Response

Reply in the enclosed file

Reviewer 3 Report

Comments and Suggestions for Authors

The authors have performed multilocus sequence typing (MLST), serotyping and antibiotic resistance screening for 31 strains of Streptococcus pneumoniae isolated from patients with invasive diseases.

In its current state, the study is only of potential relevance and of low informativeness to the microbiology community because it represents screening work on a relatively small number of samples. The study lacks identification of a link between novel strains and likely novel genetic determinants of antibiotic resistance, which would be in line with the topic of the corresponding Special Issue.

Major concerns

I strongly recommend complete serotyping of the two new strains with serotypes 8-ST53 and 19A if not already done.

With the small number of samples in operation, it is necessary to have some in-depth analysis data. The genes responsible for antibiotic resistance are well known. Thus, for the new strains with serotypes 8-ST53 and 19A, it is necessary to amplify these genes and sequence them (you can select the most interesting genes related to antibiotic resistance of your stains) and compare them with known genes and mutations in databases and publications. Alternatively, if possible, you can sequence the genomes of these two (or at least one) new strains and systematically characterize the new mutations in genes associated with antibiotic resistance. It is important not only to detect new mutations, but also to show (using bioinformatic tools) how these mutations can be linked to resistance to the corresponding antibiotic(s).

Given the low number of your samples the work need to have some data of in-depth analysis. Generally, genes responsible for resistance to antibiotics are known. So, for novel strains with serotypes 8-ST53 and 19A you need to amplify those genes and sequence them (you can choose most interesting genes related to antibiotic resistance of your stains) and compare with known genes and mutations in the databases and publications. If it is possible, you can sequence the genomes of these two novel strains and systematically characterize the novel mutations in the genes related to antibiotic resistance.

Minor comments.

Please, add countries of manufactures of reagents and laboratory equipment.

In section 2.1. please clearly indicate whether you have performed complete serotyping and if so, please indicate which antibodies were used (manufacturers, antibody dilutions, etc.).

Please add the date of access to all web pages mentioned in the text.

Figures 2 and 3. Please increase the font size.

Comments on the Quality of English Language

English is very good

Author Response

Reply in the enclosed file

Round 2

Reviewer 1 Report

Comments and Suggestions for Authors

The authors almost addressed the comments. 

Reviewer 3 Report

Comments and Suggestions for Authors

The authors have improved the manuscript. I have no further significant comments and suggest that the revised version of the manuscript be accepted for publication.